# Sex Differences in Therapies against Myocardial Ischemia-Reperfusion Injury: From Basic Science to Clinical Perspectives

**DOI:** 10.3390/cells12162077

**Published:** 2023-08-16

**Authors:** Lejla Medzikovic, Tara Azem, Wasila Sun, Parmis Rejali, Leana Esdin, Shadie Rahman, Ateyeh Dehghanitafti, Laila Aryan, Mansoureh Eghbali

**Affiliations:** Department of Anesthesiology & Perioperative Medicine, Division of Molecular Medicine, David Geffen School of Medicine, University of California Los Angeles, 10833 Le Conte Ave, CHS BH-550 CHS, Los Angeles, CA 90095, USAsunwasila@gmail.com (W.S.);

**Keywords:** drugs, myocardial infarction, ischemia reperfusion injury, sex differences

## Abstract

Mortality from myocardial infarction (MI) has declined over recent decades, which could be attributed in large part to improved treatment methods. Early reperfusion is the cornerstone of current MI treatment. However, reoxygenation via restored blood flow induces further damage to the myocardium, leading to ischemia-reperfusion injury (IRI). While experimental studies overwhelmingly demonstrate that females experience greater functional recovery from MI and decreased severity in the underlying pathophysiological mechanisms, the outcomes of MI with subsequent reperfusion therapy, which is the clinical correlate of myocardial IRI, are generally poorer for women compared with men. Distressingly, women are also reported to benefit less from current guideline-based therapies compared with men. These seemingly contradicting outcomes between experimental and clinical studies show a need for further investigation of sex-based differences in disease pathophysiology, treatment response, and a sex-specific approach in the development of novel therapeutic methods against myocardial IRI. In this literature review, we summarize the current knowledge on sex differences in the underlying pathophysiological mechanisms of myocardial IRI, including the roles of sex hormones and sex chromosomes. Furthermore, we address sex differences in pharmacokinetics, pharmacodynamics, and pharmacogenetics of current drugs prescribed to limit myocardial IRI. Lastly, we highlight ongoing clinical trials assessing novel pharmacological treatments against myocardial IRI and sex differences that may underlie the efficacy of these new therapeutic approaches.

## 1. Introduction

Cardiovascular diseases (CVDs) remain the most prevalent cause of death worldwide [1]. However, for a long time, the risk of CVD has been underestimated among women, as it was historically seen as a health condition predominantly impacting men. Coronary artery disease (CAD), which is the leading cause of myocardial infarction (MI), accounts for about 50% of CVD [2]. While a higher incidence of MI was observed in younger age groups among men compared with women [3,4], this gap was reported to narrow with increasing age [5]. Early and rapid reperfusion of affected coronary arteries with percutaneous coronary intervention (PCI) is the cornerstone of current MI treatment. However, reoxygenation via restored blood flow may induce further damage to the myocardium, leading to ischemia-reperfusion injury (IRI) [6]. Indeed, experimental studies showed that up to half of the ultimate infarct size may be due to IRI rather than the initial ischemic incident [6]. Outcomes of MI with subsequent reperfusion therapy, which is the clinical correlate of myocardial IRI, are generally poorer for women, particularly in the short term, which translates into higher in-hospital mortality compared with men [7,8,9,10].

Overall, the mortality rate for MI has declined over recent decades, which can be attributed in large part to the rapid progress in the development of new, improved treatment methods [11]. However, evidence suggests that decreases in mortality have been slower among women and have stagnated particularly among young women [12]. While studies consistently show worse outcomes for MI among women [8,9,13,14] and the benefits of current guideline-based therapies are less clear in female patients, the underlying reasons are poorly understood. New pharmacological strategies to limit myocardial IRI are the subject of numerous past and ongoing experimental studies and clinical trials [15,16]. It is becoming increasingly clear that the pharmacokinetics, pharmacodynamics, and pharmacogenetics of several drugs, including those prescribed to limit IRI, are subject to sex differences. Together with reported sex differences in disease pathophysiology, clinical presentation, and utilization of guideline-based recommended care [17], this indicates a need for further investigation of sex-based differences in treatment response and a sex-specific approach in the development of novel treatment methods. 

In this literature review, we summarize current knowledge on sex differences in the underlying mechanisms of myocardial IRI, as well as sex differences in responses to the current pharmacological treatment of myocardial IRI. Furthermore, we highlight possible sex differences in ongoing clinical trials assessing novel therapeutic drug strategies against myocardial IRI.

## 2. Pathophysiological Sex Differences in Myocardial IRI

MI most often arises from ruptured atherosclerotic plaques, resulting in acute thrombotic occlusion of coronary arteries and leading to ischemia in the myocardium [18]. Ischemia causes intracellular ATP levels to drop and calcium to accumulate in the cell [19,20]. Reperfusion restores the oxygen supply to cardiomyocytes; however, reoxygenating mitochondria take in built-up cytosolic calcium ions [19]. As a result of increasing the mitochondrial calcium levels, the mitochondrial membrane potential dissipates and enables non-selective mitochondrial permeability transition pores (mPTPs) to open [19]. Not long after the opening of mPTPs, ATP production halts, the mitochondria swells, and cytochrome C proteins abandon the membrane, ultimately causing apoptosis [1]. Damaged mitochondria also produce excessive reactive oxygen species (ROS), which are central in the induction of apoptotic and necrotic cardiomyocyte death [19]. Injured and dying cardiomyocytes release a plethora of substances that act as danger-associated molecular patterns. These patterns activate the innate immune system and trigger a pro-inflammatory response characterized by the production of pro-inflammatory cytokines and the recruitment of neutrophils and pro-inflammatory monocytes [21]. While excessive inflammation causes further tissue damage, early activation of the pro-inflammatory response is necessary for the transition to reparative responses, where anti-inflammatory monocytes predominate [21]. Additionally, fibroblasts proliferate and differentiate into myofibroblasts, depositing an extracellular matrix to maintain the structural integrity of the infarcted myocardium [21]. Myocardial IRI also promotes metabolic dysfunction in the myocardium [20]. During ischemia, the oxygen shortage suppresses the oxidative metabolism of fatty acids and other substrates and activates anaerobic glycolysis. Reperfusion washes out ischemic metabolites and supplies new oxygen, leading to a sudden start-up of oxidative metabolism leading to aerobic glycolysis [20]. These metabolic switches have a large effect on the myocardial IRI outcome. 

Existing experimental models overwhelmingly demonstrate that females not only experience greater functional recovery from MI and decreased infarct size but also the decreased outcome severity of the canonical pathophysiological mechanisms of myocardial IRI [22,23,24,25] (Figure 1). However, there is no singular explanation for the disparities in IRI severity between sexes. Experimental studies proposed intersectional origins.

### 2.1. Sex Differences in Experimental Models of Myocardial IRI

Sex differences in infarct size are well-studied in various animal models and the results point to varying magnitudes of the same conclusion: infarct size is significantly smaller in female animal models, from a conservative 25% in some studies to a nearly two-fold reduction in infarct size compared with their male counterparts [26,27,28,29,30]. The ischemic myocardium in females can be characterized by an increased tolerance to IRI, decreased postischemic contractile dysfunction, and limited fibrotic remodeling compared with the ischemic myocardium in males [23,28]. Furthermore, female mice exhibited less myofibroblast differentiation and collagen production in infarcted areas than male mice [23]. Additionally, upon MI, female mice exhibited lower levels of infiltrating pro-inflammatory monocytes and higher levels of anti-inflammatory monocytes in the infarct zone, while pro-inflammatory cytokine production was lower in the females in the spleen, but not the myocardium [23].

The mitochondria also display sexual dimorphisms. Lower numbers of mitochondria are reported to be present in cardiomyocytes from female rats; however, female mitochondria have higher oxidative capacity than mitochondria in male cardiomyocytes [31]. Rat and mouse models show that compared with males, female myocardial mitochondria undergo a greater number of post-translational modifications on enzymes that regulate ROS production and oxidative phosphorylation, have lower rates of Ca^2+^ uptake, and have more efficient recovery times for mitochondrial membrane potential [24,25]. Additionally, female mitochondria display increased Ca^2+^ transport regulation with L-type calcium channels, ryanodine calcium-release channels, and Na^+^/Ca^2+^ exchange proteins [24,25]. The female myocardium was also shown to be more tolerant to ischemia because of its relatively enhanced resistance to mitochondrial swelling, in both the rate and magnitude, at high Ca^2+^ concentrations compared with the male myocardium [24,25]. The resistance to calcium-ion-induced swelling confers at least partial attenuation of mPTP opening in response to high Ca^2+^ concentrations. Finally, lessened IRI severity specifically in female animals may also be attributed to sex differences in phosphorylation patterns that impact ROS production and elimination [32]. Augmentation in enzymatic activity of aldehyde dehydrogenase-2, which detoxes ROS aldehyde products, and phosphorylation of alpha-ketoglutarate dehydrogenase, which reduces ROS overproduction, contributes to the presentation of decreased ROS concentrations in female myocytes following IRI [32].

The delicate balance between autophagy and apoptosis after MI is also subject to sex differences. Autophagy, as a physiological foil to apoptosis, degrades nonfunctional cytoplasmic proteins and organelles, such as dysfunctional mitochondria with continuously open mPTP channels. As such, increased autophagy was shown to reduce cardiomyocyte damage [33]. After MI, levels of several anti-apoptotic proteins, including X-linked inhibitor of apoptosis protein, B cell lymphoma-extra large, and activated recruited cofactor, are decreased in male rats, while levels of these anti-apoptotic proteins remain stable pre- and post-MI in females [33,34]. Simultaneously, pro-apoptotic markers, including Bax and phospho-p38, are significantly increased in male rats post-MI compared with females [33]. On the other hand, cardiac autophagy, which is measured by the LC3B/LC3A ratio, significantly increases in female rats post-MI [33]. 

### 2.2. Roles of Sex Hormones

The female-dominant ischemic tolerance seems to carry the most weight before menopause, as postmenopausal women experience increased ROS production, chronic systemic inflammation, metabolic disorder, CAD diagnosis, and mortality from ischemic heart disease [35]. As such, sex hormones, and in particular estrogen (17β-estradiol; E2), may regulate myocardial IRI. E2 exerts genomic actions by binding the classical estrogen receptors (ERs) ERα and ERβ. Alternatively, E2 may activate membrane-bound ERα and ERβ or the G-protein-coupled receptor GPR30 (G-protein-coupled estrogen receptor) to exert rapid non-genomic effects [36]. In several rodent studies, both the pre-ischemic and post-ischemic administration of E2 was shown to result in smaller infarct size, downregulation of cardiac inflammation markers, improved heart rate, left ventricular end-systolic pressure, and ejection fraction measurements during reperfusion [35]. Administration of E2 during reperfusion yielded similar benefits, which include conserved coronary artery flow [37]. In contrast, gonadectomized female animals experienced reduced LV function and larger infarct size than control females [30,37]. 

E2 and ERs were found to regulate calcium ion levels and, consequently, mitochondrial permeability. In female cardiomyocytes, E2 was found to subvert intracellular calcium overload during reoxygenation after hypoxia [37,38,39]. Additionally, E2 was found to influence mitochondrial tolerance to rapid Ca^2+^ influxes by increasing Na^+^-dependent Ca^2+^ efflux at high E2 concentrations while increasing anti-apoptotic Bcl-2 proteins [40]. E2 mainly exerts protective effects on mitochondrial function via ERα and GPR30 [36]. GPR30 activation was found to cause cardioprotection during IRI by inhibiting the opening of the mPTP, as well as preserving mitochondrial integrity and reducing ROS production and mitophagy [41,42]. Enrichment of mitochondria-related genes was observed in cardiomyocytes from female GPR30 knockout mice versus WT mice, but not in male mice [43]. Additionally, E2, via mitochondrial p38β kinase activation, is involved in the upregulation of manganese superoxide dismutase, which is an enzyme that can reduce ROS generation, attenuate apoptosis, and diminish left ventricular infarct size when overexpressed during reperfusion [44]. In accordance with these findings, the p38β isoform was shown to activate pro-survival signaling pathways during cardiac ischemia [45]. Furthermore, E2 decreases cardiomyocyte apoptosis by increasing baseline expression of ARC levels and lowering the ratio of pro-apoptotic Bax to anti-apoptotic Bcl-2 gene expression in female rabbits compared with males [30].

Ovariectomy in rats was shown to promote cardiac inflammatory cell infiltration upon myocardial IRI, which was inhibited by E2 [39]. The production of proinflammatory cytokines tumor necrosis factor α and Interleukins-1β and -6 was shown to be inhibited by E2 and to improve cardiac function after IRI [46,47]. E2 exerted these effects via p38 MAPK, as well as GPR30. Interestingly, the enrichment of inflammatory genes was observed in cardiomyocytes from male GPR30 knockout mice versus WT mice, but not in female mice [43].

E2 signaling also plays multiple roles in cardiac metabolism [48]. It was demonstrated in rats that ERα is required to maintain cardiac glucose uptake [49]. Furthermore, E2 promotes mitochondrial basal respiration, ATP production, and respiration capacity in cardiomyocytes [50]. E2 also regulates cardiac lipid metabolism. E2 promotes the expression of Lipoprotein lipase, which is an enzyme that degrades triglycerides, in mouse hearts via direct genomic interaction through ERα and ERβ [51]. Myosin regulatory light chain interacting protein, which also functions as an inducible degrader of the LDL receptor, was induced by E2 in explanted cardiac tissue of male human donors, but not in cardiac tissue from female donors [52]. Lastly, E2 skews cardiac arachidonic acid to be metabolized to epoxyeicosanoids in women, while androgens skew arachidonic acid metabolism to hydroxyeicosatetraenoic acid (HETE) in males [48]. HETEs were recently shown to aggravate myocardial IRI in mice by promoting cardiomyocyte apoptosis and ferroptosis [53,54].

While ovariectomized animals are currently the most widely used model for menopause-related studies, it is thought that the 4-vinylcyclohexene diepoxide (VCD) mouse, which is a follicle-depleted ovary-intact model, more closely resembles the natural progression through perimenopause and postmenopause in humans [55]. A recent study showed that the hearts of menopausal mice were more sensitive to myocardial IRI than the hearts of premenopausal or perimenopausal mice, as the infarct size was approximately twofold larger during menopause [56]. No difference in infarct size was observed in pre- and perimenopausal animals [56].

Despite promising experimental studies, some clinical studies showed that estrogen replacement therapy after menopause may have varying effects on cardiovascular outcomes. Indeed, estrogen replacement therapy has been associated with a greater incidence of CAD events and thromboembolisms in postmenopausal women, but also with a lower frequency of atrial fibrillation after an MI, lower LDL, and higher HDL in postmenopausal women with previously diagnosed coronary disease [35,36]. Additionally, estrogen replacement was also reported to not significantly impact cardiovascular mortality or reinfarction in postmenopausal women with a history of MI [35,36]. One explanation for these varying outcomes is the timing of estrogen therapy initiation with respect to the start of menopause. The critical window of hormone therapy [57] hypothesizes that hormone replacement therapy is more effective when started early after menopause [58]. 

Compared with estrogens, less attention has been directed to the effect of androgens on myocardial IRI. Testosterone seems to have paradoxical effects on myocardial IRI [59]. It was found that testosterone can promote cardiac rupture, infarct size and expansion rate, myocardial neutrophil infiltration, proinflammatory cytokine production, and LV dysfunction after an MI in rodents [29,60]. These pathological parameters were demonstrated to be significantly enhanced in gonad-intact males compared with castrated controls, as well as in testosterone-treated females versus untreated controls [34]. Furthermore, in rats upon IRI, testosterone promoted the upregulation of the androgen receptor in the heart, as well as the receptor for advanced glycation end products, and downregulated anti-apoptotic Bcl-xL, resulting in a skewed balance between autophagy and apoptosis, thus aggravating IRI [29]. In contrast, other studies showed that infarct size, contractility, and mitochondrial function were impaired in castrated rats upon IRI compared with gonad-intact rats and that testosterone replacement attenuated these impairments [61]. Interestingly, testosterone treatment improved cardiac contractile function and reduced the infarct size in a more pronounced manner in aged male rats, while having no effects on the hearts of young adult rats, indicating a role of aging [62]. These conflicting experimental reports reflect clinical data wherein patients with prostate cancer undergoing androgen deprivation therapy exhibited enhanced incident MI and shorter times to fatal MI, while other studies showed no significant effects of androgen deprivation therapy [63]. In contrast, testosterone therapy in men with low testosterone seemed to increase the risk of MI [64]. However, recent studies disproved these results [65,66].

It was hypothesized that besides estrogen depletion itself, an imbalance between estrogens and androgens during menopause may underlie the loss of ischemic tolerance in postmenopausal women. The estrogen concentration decreases abruptly in the postmenopausal period, while the concentration of androgen steadily decreases over time. This leads to an increased androgen/estrogen ratio [67]. Indeed, it was reported that in postmenopausal women, a higher testosterone/estradiol ratio was associated with an elevated risk of incident CVD, CHD, and HF events [68,69]. 

### 2.3. Roles of Sex Chromosomes

The sex chromosome complement also impacts IRI outcome, independent of sex hormone differences [70]. An invaluable study tool is the four core genotypes (FCGs) mouse model wherein gonad development is independent of the sex chromosomes, thus generating four sex genotypes: XX and XY mice with either ovaries or testes [70]. XX mice were shown to be more vulnerable to myocardial IRI than XY mice, who had a markedly higher capacity for mitochondrial calcium ions, smaller infarct size, and more comprehensive recovery of cardiovascular function [71]. Gonadally identified male mice with two X chromosomes had a significantly larger infarct size and less effective recovery from MI than male mice with only one X chromosome [71]. While the exact pathophysiological mechanism remains understudied, the presence of two X chromosomes, independent of the presence of a Y chromosome, leads to larger infarct sizes after IRI [71]. One explanation may be the process of X-inactivation [72]. While theoretically transcriptional inactivation of one X chromosome is done to balance the number of genes between XX and XY cells, genomic data shows that X-inactivation is incomplete [70]. Nearly 20% of X chromosome genes elude inactivation and have augmented expression in XX individuals compared with their XY counterparts. These X escapee genes may affect myocardial IRI and were shown to comprise, among others, kdm5c/kdm6a, which regulate pro-inflammatory cytokine production [73]; usp9x, which is implicated in cell death pathways; and bmx and sts, which promote fibroblast activation [74].

## 3. Sex Differences in Current MI Pharmacological Therapies

It is yet unclear whether sex differences in MI outcomes and mortality in particular are a result of treatment disparities and dissimilarities in clinical characteristics, such as comorbidities between men and women, or related to sex-based biological and pathophysiological distinctions, leading to different treatment responses (Figure 1). It is likely that each of these factors contributes to sex differences in MI outcomes to a certain degree.

### 3.1. Sex Differences in Drug Responses

The pharmacokinetics, pharmacodynamics, and pharmacogenetics of several drugs are subject to sex differences. This includes cardiovascular drugs that are prescribed after an MI to preserve cardiac function, such as β-blockers, renin–angiotensin–aldosterone inhibitors, and antithrombotics, which have become a cornerstone in the pharmacological treatment for acute MI [75]. 

Physiological differences between males and females may affect drug metabolism. Men exhibit faster absorption, processing, and excretion of most drugs compared with women. Furthermore, differences exist in the accumulation and distribution of hydrophilic and lipophilic drugs between the sexes, as women tend to have a higher percentage of body fat and lower plasma volume [76]. A study of 86 different FDA-approved drugs revealed that 76 of those exhibited prolonged elimination times and increased blood concentrations in women [77]. Of those drugs, 96% were associated with higher incidences of adverse drug reactions (ADRs) in women [77]. Indeed, women tend to experience ADRs more frequently than men. For example, women report more ADRs from diuretics, ACE inhibitors, anticoagulants, statins, and antiarrhythmic drugs [78]. Women are reported to have an increased risk of severe bleeding from Aspirin, GPIIb/IIIa inhibitors, anticoagulants, and antiplatelet drugs [76,79]. The enhanced reduction in heart rate and blood pressure due to β-blockers is more common in women compared with men, as well as higher rates of hypo-osmolarity, hypokalaemia, and hyponatremia from diuretics [79].

Women are reported to have higher gastric pH and a lower small intestinal fluid volume [76]. As such, drugs that require an acidic environment for absorption, such as the β-blocker Metoprolol or the calcium channel blocker Verapamil, may have lower oral bioavailability in women, and formulations intended for duodenal absorption, including enteric-coated aspirin, may show reduced or delayed absorption in women [79]. Glomerular filtration rate and tubular secretion, which are important factors in the renal clearance of drugs, are reported to be lower in women than in men [80]. Indeed, Verapamil, Metoprolol, and Propranolol exhibit slower renal clearance in women [78]. The hepatic metabolism also differs between the sexes, especially the cytochrome P450 (CYP) family of catalyzing enzymes. Higher CYP1A2, 2D6, and 2E1 activities in men were reported by multiple studies, while CYP3A4 and 2B6 activities are higher in women [76]. Up to 50% of drugs currently used are CYP3A4 substrates and women are reported to exhibit ~25% higher clearance of these drugs than men [81]. Furthermore, endogenous estrogen, as well as oral contraceptives, interact with CYP450 enzymes. Both estrogen and progesterone compete with drugs for degradation via CYP450 enzymes. Additionally, steroid hormones activate the expression of CYP3A4, and ERa is reported to modulate CYP1B1 expression [82,83]. In the intestine, the efflux membrane transporter P-glycoprotein (P-gp) extrudes and limits the cellular uptake of toxins and xenobiotics [84,85]. Several cardiovascular drugs, including the β-blockers Labetalol and Propranolol, the calcium channel blocker Verapamil, the ARB inhibitor Losartan, and the antiplatelet Ticagrelor, are transported by P-gp but also act as a P-gp inhibitor, thus achieving enhanced drug bioavailability and uptake [84,85]. P-gp expression is reported to be higher in men and to be sensitive to modulation by both estrogen and testosterone [80,86]. 

### 3.2. Sex Differences in Clinical Characteristics

Multiple reports suggest differential clinical characteristics between male and female MI patients with a higher prevalence of comorbidities and cardiovascular risk factors being observed among female patients [10,14,87]. In addition, presentation with atypical symptoms of MI other than classic chest pain was reported more commonly in women, which may lead to an increased risk for misinterpretation of their clinical symptoms by healthcare providers [88].

Major bleeding complications after a PCI are one factor contributing to worse outcomes among female patients [89] and studies consistently linked the female sex with an elevated risk of bleeding and vascular complications during a PCI [13]. This highlights the importance of taking into account significant biological differences, such as smaller vessel size and higher prevalence of MI with non-obstructive coronary arteries (MINOCA) among women, which may limit the therapeutic benefits of PCI in some cases [90]. These pathophysiological and biological differences may also impact the efficacy of standard drugs for the treatment of MI. A previous report on the lack of significant improvement in long-term clinical outcomes with dual antiplatelet therapy in MINOCA patients serves as one example [91]. Furthermore, while heparin is the recommended drug of choice for peri-procedural and adjunctive anticoagulation to reperfusion therapy [92], evidence suggests that anticoagulation with unfractionated heparin poses a particularly high bleeding risk among women. Even when administered weight-adjusted doses, women undergoing treatment with unfractionated heparin for MI experience greater activation of partial thromboplastin time than men, putting them at a greater risk for bleeding complications [93]. This increased sensitivity to heparin may be a reason to opt for a different anticoagulation strategy in female patients. Subgroup analysis of the VALIDATE-SWEDEHEART trial showed a significant reduction in major bleeding events after a PCI in female MI patients who received Bivalirudin as an anticoagulant medication instead of unfractionated heparin, while no significant benefit was observed among male patients [94].

### 3.3. Sex Differences in Standard of Care

It is notable that the standard of care differs between male and female MI patients, partly due to the less timely coronary reperfusion through PCI and a longer ischemic time among women [95,96]. Aside from this delay in treatment, women are also less likely to even receive coronary reperfusion therapy, either in the form of pharmacological fibrinolysis or invasive coronary revascularization via PCI [97]. Furthermore, previous studies showed that the likelihood of receiving early Aspirin and β-blocker treatment was lower among women. While receiving diuretics more often, prescriptions for β-blockers; antiplatelet drugs, such as Aspirin; and potent P2Y12 inhibitors, ACE inhibitors, and statins were less frequent for female compared with male patients during hospitalization [10,98,99]. In addition, significant sex disparities exist in secondary prevention in MI patients, as the rates of referral, enrollment, and completion of cardiac rehabilitation are lower in women compared with men [100,101]. 

These reports indicate that women that present with MI are less likely to receive standard evidence-based care in accordance with clinical guideline recommendations.

## 4. Ongoing Clinical Trials for Drugs against Myocardial IRI: Potential Sex Differences

Numerous clinical trials are currently ongoing testing the efficacy of new drug interventions against myocardial IRI. On www.clinicaltrials.gov (17 April 2023), the filters ‘myocardial infarction’, ‘reperfusion injury’, ‘myocardial’, ‘status recruiting’, ‘age 18–65+’, and ‘intervention: drug’ yield several trials targeting oxidative stress, inflammation, thrombosis, and lipid and glucose metabolism in myocardial IRI. Potential sex differences may underlie the efficacy of these interventions (Table 1).

### 4.1. Drugs Targeting Oxidative Stress

FDY-5301 is a drug that contains sodium iodide, which is an inorganic halide and elemental reducing agent that may act as a catalytic anti-peroxidant in the conversion of hydrogen peroxide to water and oxygen [155]. A pilot study showed that i.v. delivery of FDY-5301 in STEMI patients before re-opening the occluded artery led to decreased infarct size and higher LVEF 3 months after an MI. Although the study was not powered to detect statistical significance in cardiac function, FDY-5301-treated patients exhibited significantly lower plasma levels of myeloperoxidase and NTproBNP [155]. The ongoing trial NCT04837001, which is a larger phase III trial, aims to validate these promising pilot results and monitor cardiovascular mortality or acute heart failure over 12 months. Hydrogen peroxide is a relatively stable ROS formed by the antioxidant enzyme superoxide dismutase and hydrogen peroxide is further metabolized to water and oxygen by either catalase or glutathione peroxidase. Superoxide dismutase activity in the hearts of female rats was found to be higher than in male rats; however, a gonadectomy led to a decrease in superoxide dismutase levels in both sexes [102]. Catalase activity was found to be higher only in the kidneys of female vs. male rats and GPx activity was demonstrated to be lower overall in females compared with males [102,103,104]. However, other studies did not find sex differences in the expression or activity levels of antioxidant enzymes [102,103,104]. Interestingly, it was hypothesized that the greatest difference in antioxidant properties between the sexes is likely due to estrogen. Estrogen contains a phenolic hydroxyl group, which has free radical scavenging activity [102]. Furthermore, E2 can promote superoxide dismutase and glutathione peroxidase gene expression by activating MAPK and NFkB signaling [105]. Whether the anti-peroxidant action of FDY-5301 in STEMI is affected by estrogen remains to be elucidated.

The cardioprotective effects of adenosine in myocardial IRI have been well established and range from reductions in oxidative stress, vasodilatation, and anti-inflammatory properties, to the regulation of calcium homeostasis [156,157,158,159,160]. Previous studies showed a link between adenosine treatment and a reduction in the infarct size of ischemic and subsequently reperfused myocardium in animal models [161], which prompted the launch of numerous clinical trials. However, conflicting results were obtained regarding post-MI cardiac function, which is likely explained by dissimilarities in the conditions of adenosine administration, particularly regarding dose, time, and duration of administration [161]. A clinical trial, namely, NCT05014061, in STEMI patients evaluating the impact of intravenous adenosine infusion initiated prior to revascularization on the reversal of myocardial stunning and on cardiac function after 48 h is ongoing. Aside from optimizing the conditions of adenosine administration, it is also necessary to take biological sex differences that may influence the therapeutic efficacy into account. A previous study on the cardioprotective effects of cardiac adenosine A1 receptor activation in male and female mice revealed that A1 receptor stimulation led to increased endothelial nitric oxide synthase phosphorylation through the phosphoinositide 3-kinase/protein kinase B/eNOS signaling axis in male but not in female hearts [162]. However, eNOS phosphorylation in female hearts was higher at baseline and a prior report by the same group established a link between higher eNOS phosphorylation and cardioprotection against myocardial IRI particularly in females [107]. These results suggest a possible redundancy between adenosine-induced cardioprotection and innate cardioprotective mechanisms in females, calling into question the added therapeutic benefit of adenosine for female patients. On the other hand, the possibility of different cardioprotective adenosine signaling pathways between males and females should also be considered.

### 4.2. Drugs Targeting Inflammation

Inflammation and healing in myocardial IRI are closely connected via matrix metalloproteinases (MMPs), which are a family of proteolytic enzymes that regulate extracellular matrix turnover and inflammatory signaling [162]. After an MI, pro-inflammatory cytokines enhance MMP expression, especially MMP-2 and -9, to facilitate ECM degradation, inflammatory cell recruitment, and cytokine processing. However, sustained MMP activation leads to adverse remodeling of the myocardium. As shown in both patient and animal models, MMP-2 levels in the heart, as well as plasma, significantly increase within the first 24 h after an MI and upon reperfusion therapy [163,164]. MMP-2 expression and activity are subject to sex-related differences [165]. The activity of serum pro-MMP-2 was found to be decreased in women suffering from HF compared with men [108]. In mice, higher MMP-2 activity was observed in the LV of males vs. females after an MI [166]. Sex hormones seem to exert dual effects on MMP-2 in the heart. MMP-2 expression is downregulated in volume-overloaded hearts of ovariectomized rats, while enhanced MMP-2 activity was reported in isolated healthy hearts of ovariectomized rats [109]. E2 treatment was also shown to inhibit MMP-2 transcription and expression in rat cardiac fibroblasts and cardiac inflammatory cells [110,111,112]. Testosterone also seems to affect cardiac MMP-2 expression, as castration in male rats significantly reduced the MMP-2 expression in volume-overloaded hearts [113]. Doxycycline is the only inhibitor of MMP-2 approved for clinical use. Doxycycline is commonly used to treat rosacea and periodontitis, and at higher doses, acts as an antibiotic. Previously, the TIPTOP trial found that orally administering Doxycycline immediately after a PCI and then for 7 days after resulted in improved cardiac remodeling indices at 6 months post-MI, together with altered levels of MMP-2 and tissue inhibitor of metalloproteinases-2 (TIMP-2) [167]. The double-blinded phase II trial NCT03508232 is ongoing to confirm these promising pilot results. 

In addition to the innate immune system, the adaptive immune system also plays a role in the immune response after an MI [168]. Experimental studies showed that B cells are instrumental in orchestrating the inflammatory response after myocardial IRI, partially via mobilizing inflammatory monocytes to the infarct site [169]. B cell depletion in mice using a single dose of anti-CD20 antibody after MI led to reduced recruitment of pro-inflammatory monocytes and reduced infarct size with improved cardiac function [169]. Rituximab is a monoclonal antibody that targets the B cell surface protein CD20. NCT05211401 is currently enrolling to compare the effect of a single injection of two doses of Rituximab infused within 3 h of a PCI on left ventricular systolic function after 6 months in patients who have had an acute STEMI. A pilot study showed that Rituximab was safe and able to deplete circulating B cells in STEMI patients [170]. Rituximab is currently used to treat certain autoimmune diseases and cancers. Male patients with diffuse large cell lymphoma had a higher clearance of Rituximab than female patients, together with poorer treatment outcomes and shorter progression-free survival in males than in female patients treated with Rituximab [114]. Similar results were obtained in follicular lymphoma patients treated with Rituximab [115]. In membranous nephropathy, female patients were found to be more resilient to renal injury and achieve complete or partial remission following Rituximab therapy compared with men [116]. Taken together, Rituximab therapy may be more efficient in women than men. Interestingly, single-cell sequencing of peripheral immune cells sampled from men and women showed that females exhibited higher B-cell-activated signaling at the transcriptional level than males already at baseline healthy conditions [117].

Glucocorticoids are potent regulators of the inflammatory response with a wide range of actions. In the acute inflammatory phase, glucocorticoids suppress cytokine and chemokine production and reduce leukocyte infiltration and leukocyte binding to endothelial cells [171]. In the resolution phase of inflammation, glucocorticosteroids may promote the clearance of apoptotic cells and promote anti-inflammatory phenotypes in macrophages; however, glucocorticoids also reduce collagen synthesis in fibroblasts [171]. Glucocorticoids act by binding to the ubiquitously expressed glucocorticoid receptor, as well as the mineralocorticoid receptor, and also exert non-genomic actions [172]. The non-genomic pathway employed by glucocorticoids is induced rapidly after administration. Indeed, glucocorticoids exert rapid effects on ROS production by modulating NO synthase; on intracellular calcium levels by modulating SERCA2A, adenylyl cyclase, and protein kinase A activity; and on inflammation and apoptosis by acting on MAPKs, cAMP levels, and mitochondrial function [172,173]. As such, trial NCT05462730 is currently enrolling to assess whether a single bolus of methylprednisolone infusion administered in the prehospital setting prior to PCI can limit reperfusion injury and reduce the final infarct size 3 months after STEMI. However, the response to glucocorticoids may be influenced by sex-specific pharmacokinetics and pharmacodynamics, as well as sex hormones. The clearance of methylprednisolone was reported to be higher in women compared with men [118]. The sensitivity of basophil trafficking to methylprednisolone treatment was found to be related to plasma estradiol in women [118]. However, in male rats, methylprednisolone clearance was measured to be threefold higher in males compared with females, regardless of the estrous phase in the females [119]. A sepsis model in rats demonstrated that sex hormones affect the anti-inflammatory actions of glucocorticoids, as there was no difference in survival between castrated and gonad-intact endotoxemic rats treated with dexamethasone. In contrast, the survival rate of female endotoxemic rats treated with dexamethasone was increased after an ovariectomy [120]. These data are in line with reports of adjunctive hydrocortisone treatment in patients with septic shock leading to a significant decrease in the length of mechanical ventilation and ICU stay only in males [121]. Interestingly, dexamethasone treatment in rats revealed sex-specific glucocorticoid-regulated gene expression in the liver, which is a classic glucocorticoid-responsive organ, in several canonical pathways, such as apoptosis signaling, hypoxia signaling, and interferon and interleukin signaling [120]. Indeed, dexamethasone treatment expands the number of sex-dimorphic genes expressed in the liver compared with untreated rats.

### 4.3. Drugs Targeting Thrombosis

Thrombus formation is a hallmark of occluded coronary arteries in MI, but can also manifest as in-stent thrombosis after a PCI [18]. Thrombosis is driven by platelet activation and aggregation and antiplatelet agents have become the mainstay antithrombotic therapy after an MI or PCI [174]. Platelets in damaged vessels release dense granules containing substances that cause the recruitment of circulating platelets to the injury site and further platelet activation [175]. Adenosine diphosphate is one of these substances. ADP binds the P2Y12 receptors on platelets, and downstream signaling ultimately leads to integrin αIIbβ3, also known as GPIIb/IIIa, to adopt a high-affinity state for fibrinogen and other ligands [175,176]. This promotes signaling cascades that drive key platelet functions such as spreading, aggregation, and thrombus consolidation [176]. As such, integrin αIIbβ3 and the P2Y12 receptor play a central role in platelet biology and arterial thrombosis. Zalunfiban is a novel small molecule inhibitor of the platelet αIIbβ3 receptor specifically designed for the first medical contact therapy of STEMI patients. Trial NCT04825743 is currently recruiting to assess the effects of a single weight-based dose of subcutaneous Zalunfiban administered by the ambulance staff prior to a PCI on acute stent thrombosis, MI recurrence, new-onset HF, and all-cause death 30 days after MI. The drug Ticagrelor inhibits platelet aggregation by antagonizing the P2Y12 receptor. NCT05149560 is currently assessing twice-daily treatment with oral Ticagrelor following PCI on stent thrombosis, MI recurrence, and the composite of cardiac death 3 months post-MI. While no sex-specific effects of Zalunfiban have yet been reported, treatment with other GpIIb/IIIa inhibitors is reported to result in bleeding more often in women than men [122,123]. After undergoing PCI, women are reported to benefit less from Ticagrelor treatment in terms of major adverse cardiovascular outcomes (MACE) and cardiovascular death than men and are at greater risk of major bleeding [133,134,135,136]. However, a meta-analysis of clinical trials that included a total of 63,346 men and 24,494 women with CAD reported that both the efficacy and safety of P2Y12 inhibitors appeared to be similar between the sexes [177]. Differences in function between men and women were reported from platelet counts, expression of surface receptors, and functional reactivity. Both animal models and studies in humans showed that platelets in females are more reactive than platelets in males when and that women have higher platelet counts [124,125,126,127]. However, the exact mechanism underlying sex differences in platelet biology and responses to antithrombotic therapy is yet unclear. It was reported that the menstrual cycle affects platelet reactivity, alluding to an effect caused by sex hormones [128,129]. However, conflicting reports regarding the effects of E2 on platelet activation were published, with E2 both promoting and inhibiting platelet aggregation and thrombus formation in mice [130]. Similar contradictory results were found in multiple studies on the effect of estrogen therapy on platelet activation in menopausal women [130]. Additionally, testosterone promotes aggregation of both male and female platelets equally and both low levels of circulating testosterone, as well as abuse of anabolic androgenic steroids, have been linked to thrombotic events [131,132].

The plasminogen system, which comprises tissue-type plasminogen activator (tPA), urokinase-type plasminogen activator, and their inhibitor plasminogen activator inhibitor (PAI-1), plays a major role in fibrinolysis of thrombi and tissue remodeling. Impaired fibrinolytic capacity has been associated with an increased risk of MI [177]. While PCI is deemed safer, fibrinolysis may be used in MI patients where PCI is not possible [178,179]. Clinical trials NCT03998319, NCT03335839, and NCT02894138 are assessing the effects of adjunctive intracoronary tPA administration immediately following PCI on final infarct size, rehospitalization, and cardiovascular death 1–24 months post-MI. In patients that had an acute ischemic stroke, women were reported to benefit more from therapy with recombinant tPA [138]. However, a recent study showed that 3 months after a stroke and treatment with recombinant tPA, men showed better functional outcomes than women. Sex differences exist in the fibrinolytic system [139]. In healthy middle-aged subjects, women were found to release more endothelial tPA antigen than men [140]. Female pulmonary hypertension patients suffering from thrombosis in the pulmonary vessels exhibit elevated tPA antigen levels and lower tPA levels and higher PAI-1 activity, leading to an antifibrinolytic/prothrombotic state compared with male patients [141]. In a systematic review of 24 studies, no differences in tPA levels were found between male and female stroke patients suffering from migraine, while PAI-1 was elevated in females [142]. Sex hormones may affect the fibrinolytic system, as hormonal contraceptives have long been associated with reduced tPA levels and elevated levels of thrombin-activated fibrinolysis inhibitor [180]. In contrast, E2 was recently shown to repress the expression of PAI-1 in an ER-mediated manner in vascular smooth muscle cells, resulting in enhanced tPA activity [143].

### 4.4. Drugs Targeting Lipid Metabolism

Studies on the FCG mouse model showed that cholesterol levels are higher in mice with testes than in mice with ovaries, independent of the XX or XY chromosome complement [144]. Statins inhibit cholesterol biosynthesis and lower plasma cholesterol levels by functioning as potent competitive inhibitors of 3-hydroxy-3-methylglutaryl-coenzyme A (HMG-CoA) reductase, which is the enzyme that catalyzes the rate-limiting step in cholesterol synthesis [181]. Interestingly, statin-induced reductions in cholesterol levels were larger in mice with testes than in mice with ovaries [144]. The central organ for cholesterol synthesis and action of statins is the liver. Gonadal and chromosomal sex independently affect the liver transcriptome in hypercholesterolemia, and chromosomal sex in particular seems to impact the hepatic transcriptional response to statin treatment [144]. For instance, statin treatment leads to a compensatory upregulation of HMG-CoA reductase, which may reduce statin drug efficacy, and it is suggested that this response is driven by the XY chromosome complement [144]. By lowering plasma cholesterol, statins lower the risk of CVD. Interestingly, evidence suggests that independent of their lipid-lowering effects, statins exert anti-inflammatory, antioxidant, and platelet anti-aggregation functions as well as improve endothelial function [182]. Both the pathophysiology of CAD itself, as well as angioplasty and PCI, induce platelet activation, thrombosis, and inflammation [18]. As such, several clinical trials investigated whether in addition to the long-term benefits associated with lipid-lowering, preloading with a single statin dose prior to a PCI also may play a beneficial role early after a PCI, reporting promising results. Currently, trial NCT04974814 is comparing different statin subtypes. Whether sex differences in statin anti-inflammatory and antiplatelet effects exist remains to be discovered.

Proprotein convertase subtilisin/kexin type 9 (PCSK9) is a major player in cholesterol homeostasis. PCSK9 binds to and disrupts endocytic recycling of LDL receptors to the cell surface and promotes subsequent LDLR degradation in the lysosomes [182]. This process inhibits LDL cholesterol (LDL-C) uptake by cells, leading to hypercholesterolemia. PCSK9 inhibitors added to statin therapy were found to significantly reduce LDL-C, as well as lowering MACE [183]. Sex differences in LDL-C and PCSK9 metabolism were previously reported. LDL-C levels in women increase after midlife and several real-world registries reported that in patients starting PCSK9 inhibitors, LDL-C was significantly higher in women than men [148,149,150,151]. Furthermore, levels of circulating PCSK9 were consistently shown to be higher in women versus men in several studies [145]. Estrogen is thought to regulate PCSK9 via genomic and non-genomic mechanisms. Both ERα and GPER activation were shown to repress PCSK9 expression and promoter activity [146,184]. Additionally, internalization of the PCSK9-LDLR complex was shown to be inhibited by E2 and GPR30 activation through the decreased phosphorylation of PCSK9 [146,147]. Interestingly, several recent studies reported that PCSK9 monoclonal antibodies are less effective in lowering LDL-C levels in females compared with males and that after confounder correction, sex is a significant predictor of the therapeutic response to PCSK9 antibodies [148,149,150,151]. Ongoing trials NCT05284747 and NCT04951856 aim to assess the effects of monoclonal PCSK9 antibody Evolocumab on a composite of MI, revascularization, and all-cause death when administered to MI patients before undergoing PCI and subsequently added to standard lipid management. Whether sex differences are also observed in the efficacy of PCSK9 antibody treatment against MI injury remains to be concluded from the outcomes of the ongoing clinical trials.

### 4.5. Drugs Targeting Glucose Metabolism

Several inhibitors targeting the sodium glucose co-transporter 2 (SGLT2) have been developed to treat hyperglycemia in type 2 diabetes patients. SGLT2 inhibitors act by inhibiting glucose reabsorption in the kidney’s proximal tube. Interestingly, multiple trials showed that SGLT2 inhibitors also decrease MACE in type 2 diabetes patients, however to a far lesser extent in women compared with men [185,186]. However, there are also reports of similar protection against cardiovascular events between the sexes with SGLT2 inhibitor use [187,188,189,190]. SGLT2 expression patterns may explain the observed sex differences in the clinical setting. In rats, it was shown that while SGLT2 is expressed in multiple organs, including the heart, the expression of SGLT mRNA is higher in females than in males, especially in the kidney [152,153]. In contrast, SGLT2 protein expression was reported to be higher in the kidneys of male rats [152,153]. Additionally, a hormonal upregulation of SGLT2 takes place after puberty in female rats but not in male rats [152,153]. Several trials involving HF patients showed that SGLT2 inhibitors also significantly reduce MACE regardless of diabetic status [154,188,189]. However, the exact mechanism by which SGLT2 inhibitors exert these protective effects is yet unclear. It is hypothesized that SGLT2 inhibitors promote cardiac energy metabolism, circulating progenitor cells, and erythropoiesis, while also decreasing blood pressure, inflammation, and adverse cardiac remodeling [191]. A systematic review of five clinical trials involving 21,947 HF patients showed that SGLT2 inhibitors reduce the risk of primary composite HF in both men and women, but that the benefits are less pronounced in women [154]. Trials NCT05305911, NCT04363697, and NCT04509674 are currently ongoing to assess the effects of the in-hospital initiation of SGLT2 inhibitors Dapagliflozin and Empagliflozin in addition to a standard therapeutic regimen after an MI on the outcome of HF hospitalizations, MACE, and all-cause mortality. Whether sex differences are also observed in the effects of SGLT2 inhibitors in MI patients remains to be elucidated upon the conclusion of these clinical trials.

## 5. Conclusions

Despite enormous pre-clinical efforts to develop innovative therapies against myocardial IRI and the high numbers of clinical studies that have been conducted thus far, unfortunately, only a few clinically effective therapies exist. This can be partially attributed to the complexity of the disease pathology. Processes like reoxygenation, inflammation, and fibrosis are a double-edged sword in myocardial IRI in which a balance needs to be achieved in a timely manner for an optimal healing response [21]. The many sex differences that exist in the several pathological processes involved in myocardial IRI further complicate finding an optimal treatment strategy.

Experimental models have overwhelmingly demonstrated a female-dominant tolerance against myocardial IRI. However recent studies show that women experience worsened outcomes after a PCI. There is a dire need to fill the knowledge gaps underlying this seeming disparity between experimental studies and clinical outcomes. It is distressing that women are reported to benefit less from current guideline-based therapies and that long-term follow-up studies show that outcome disparities between the sexes have not narrowed over time. Historically, women have often been excluded from clinical pharmaceutical trials, leading to knowledge gaps regarding sex-related differences. Based on current literature, it is evident that sex-based differences in efficacy and safety of pharmacotherapies for patients should be considered when investigating novel therapies for myocardial IRI. For instance, it may be necessary to reassess drug dosages and consider sex-specific adjustments since there are well-known sex differences in drug pharmacokinetics and pharmacodynamics of cardiovascular medications. Additionally, a sex-based approach to clinical trials assessing novel therapeutic strategies against an MI and subsequent IRI could possibly eliminate the sex-dimorphic outcomes in the future.

## Figures and Tables

**Figure 1 cells-12-02077-f001:**
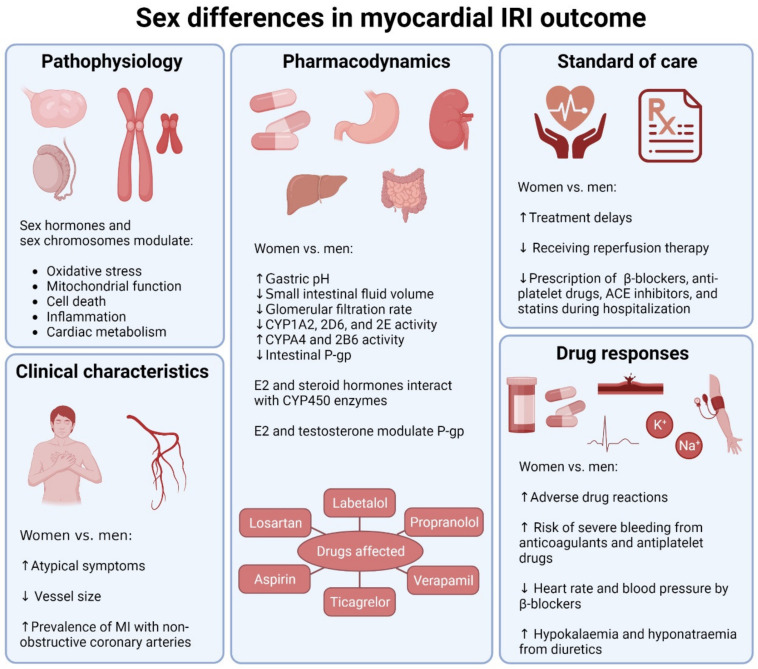
Factors contributing to sex differences in outcomes of myocardial ischemia-reperfusion injury. Sex differences underlie biological pathophysiology, clinical characteristics, pharmacodynamics of and response to cardiovascular drugs, and standard of care. ACE: angiotensin-converting enzyme, CYP: cytochrome P450, E2: 17β-estradiol, MI: myocardial infarction, P-gp: the efflux membrane transporter P-glycoprotein. Created with BioRender.com, accessed on 27 June 2023.

**Table 1 cells-12-02077-t001:** Ongoing clinical trials researching new pharmacological strategies against myocardial ischemia-reperfusion injury. E2: 17β-estradiol, eNOS: endothelial nitric oxide synthase, F: female, GPR30: G-protein-coupled estrogen receptor, GPx: glutathione peroxidase, HF: heart failure, i.v.: intravenous, ICU: intensive care unit, LDL-C: low density lipoprotein cholesterol, LV: left ventricle, M: male, MACE: major adverse cardiovascular endpoint, MMP: matric metalloprotease, PAI: plasminogen activator inhibitor, PCI: percutaneous coronary intervention, PCSK9: proprotein convertase subtilisin/kexin type 9, s.c.: subcutaneous, SGLT: sodium glucose cotransporter, SOD: superoxide dismutase, tPA: tissue plasminogen activator.

Target	Clinical Trial	Drug and Administration Route	Proposed Mechanism	Possible Sex Differences
Oxidative stress	NCT04837001	FDY-5301, i.v. delivery before a PCI	Anti-peroxidant, promotes the conversion of hydrogen peroxide to water and oxygen	↑ SOD activity in hearts of F rats [102] ↓ SOD expression after gonadectomy in both sexes [102] ↑ Catalase activity in F rat kidneys [102,103,104] ↓ GPx activity in F rodents [102,103,104] Estrogen phenolic hydroxyl group scavenges free radicals [102] E2 promotes Mn-SOD and GPx gene expression [105]
	NCT05014061	Adenosine, i.v. delivery before a PCI	Antioxidative, vasodilatation, anti-inflammatory, regulation of calcium homeostasis	↑ Cardiac adenosine A1 receptor-induced eNOS phosphorylation in M [106,107]
Inflammation	NCT03508232	Doxycycline, oral immediately after a PCI and 7 days after a PCI	MMP-2 inhibition	↓ Activity of serum pro-MMP-2 activity in F HF patients [108] ↑ LV MMP-2 activity in M mice after MI [60] ↑ Cardiac MMP-2 activity in healthy ovariectomized F rats [109] ↓ Cardiac MMP-2 expression upon volume overload in ovariectomized F rats [109] E2 inhibits MMP-2 expression in rat cardiac fibroblasts and cardiac inflammatory cells [110,111,112] ↓ Cardiac MMP-2 expression upon volume overload in castrated M rats [113]
	NCT05211401	Ritixumab, i.v. within 3 h of PCI	CD20 antibody, B cell depletion, anti-inflammatory	↑ Rituximab clearance and poorer treatment outcomes in M DLBCL patients [114,115] ↑ Complete or partial remission following Rituximab therapy in F nephropathy patients [116] ↑ Baseline B-cell-activated signaling in peripheral immune cells in F [117]
	NCT05462730	Methylprednisolone, single i.v. bolus in the prehospital setting before PCI	Glucocorticoid, anti-inflammatory, antioxidant, promoted mitochondrial function, regulation of calcium homeostasis	↑ Methylprednisolone clearance in F patients [118] ↑ Methylprednisolone clearance in M rats [119] Sensitivity of basophils to methylprednisolone treatment to plasma E2 in F [118] ↑ Survival rate in ovariectomized endotoxemic F rats treated with dexamethasone [120] Castration in M rats had no effect on dexamethasone treatment of endotoxemia [120] ↓ Length of mechanical ventilation and ICU stay by hydrocortisone treatment only in M septic shock patients [121] Dexamethasone treatment in rats promotes sex-specific glucocorticoid-regulated gene expression in the liver [120]
Thrombosis	NCT04825743	Zalunfiban, administered by the ambulance staff prior to a PCI	Platelet αIIbβ3 receptor inhibitor, inhibition of platelet aggregation	↑ Bleeding in F after GpIIb/IIIa inhibitor treatment [122,123] ↑ Platelet count and reactivity in F [124,125,126,127] Menstrual cycle affects platelet reactivity [128,129] Contradictory effects of E2 and testosterone on platelet activation [130,131,132]
	NCT05149560	Ticagrelor, oral after PCI for 12 months	Platelet P2Y12 receptor inhibitor, inhibition of platelet aggregation	↑ Bleeding in F after Ticagrelor treatment [133,134,135,136] Less pronounced benefits of Ticagrelor on MACE in F vs. M [133,134,135,136] No difference in efficacy and safety of P2Y12 inhibitors between M and F [137]
	NCT03998319, NCT03335839, and NCT02894138	Tenecteplase or Alteplase, intracoronary administration immediately following a PCI	Recombinant tPA, promotion of fibrinolysis	Less pronounced benefits of recombinant tPA in M ischemic stroke patients [138] Better functional outcomes after recombinant tPA in M ischemic stroke patients [139] ↑ Endothelial tPA antigen release in F [140] ↑ tPA antigen, ↑ PAI-1 activity, and ↓ tPA levels in pulmonary arteries of F PAH patients [141] ↑ PAI-1 in F stroke with migraine patients, no differences in tPA levels between M and F [142] E2 represses PAI-1 expression via ER in vascular smooth muscle cells [143]
Lipid metabolism	NCT04974814	Rosuvastatin and Atorvastatin, single high-dose preloading before a PCI	Anti-inflammatory, antioxidant, inhibition of platelet aggregation, improve endothelial function	↑ Statin-induced reductions in cholesterol levels in mice with testes than mice with ovaries [144] Sex chromosome complement impacts hepatic transcriptional response to statins [144]
	NCT05284747 and NCT04951856	Evolocumab, s.c. immediately after a PCI and biweekly after	PCSK9 antibody, promotion of revascularization	↑ Circulating PCSK9 in F [145] PCSK9-LDLR complex internalization inhibited by E2 and GPR30 [146,147] PCSK9 antibodies were less effective in lowering LDL-C in F [148,149,150,151]
Glucose metabolism	NCT05305911, NCT04363697, and NCT04509674	Dapagliflozin or Empagliflozin, oral after a PCI once daily for two or six months	SGLT2 inhibitor, promotion of cardiac energy metabolism, enhanced circulating progenitor cells, erythropoiesis, decreased blood pressure, anti-inflammatory	↑ SGLT2 mRNA in F rats, ↑ SGLT2 protein in M rats [152,153] Less pronounced benefits of SGLT2 inhibitors on HF risk in F vs. M [154]

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
