# Peer review of "Sex Differences in Therapies against Myocardial Ischemia-Reperfusion Injury: From Basic Science to Clinical Perspectives"

_cells, 2023, doi:10.3390/cells12162077_

Round 1
Reviewer 1 Report
Females generally enjoy a cardiovascular advantage for cardiovascular disease, in large part due to the relatively higher levels of estrogens. Despite this endogenous protection they have worse outcomes from acute myocardial infarction. The reasons for this paradox are various, ranging from biological to sociological. This literature review investigates the various explanations for the sex differences in AMI outcomes, with a focus on biological mechanisms and potential differences in the response to treatments.
The manuscript is comprehensive in its coverage and is a nice overview of many key areas. While more depth could be provided in any one of these areas, there is some advantage to the broader review which I think will be helpful for a diverse audience.
1. Studies using ovariectomized animals provide insight into estrogen deficiencies, but do not recapture the more complex changes seen in human menopause. The VCD induced ovarian failure model (reviewed by Brooks HL et al Physiology. 31(4): 250-7. 2016) is likely a better model in terms of capturing more of these changes. Some acknowledgment in the review of the limitations of the OVX models and the potential of the VCD model would be helpful to put the OVX data in context.
2. The criticisms of estrogen replacement therapy are fair and correct, but a little too definitive. It should be noted that "some" studies show negative effects, instead of the more definitive presentation. The issue of timing does address some of these limitations and I appreciate the addition, but I believe the statement still requires some softening.
3. One area that is not addressed is the more complex effects driven by an androgen-estrogen imbalance after menopause. After menopause, androgen levels increase relative to estrogens (not necessarily an absolute increase). Because of the opposing effects of these hormones, some negative effects on the CV system may be a result of an imbalance in hormones. Some notation of this possibility could be added to the review (the idea that the negative effects are not just due to estrogen loss, but also due to an androgen:estrogen imbalance).
4. Section 3 opens by pointing out that it is unclear if sex differences are driven by treatment disparities, comorbidities, or biological differences. But it doesn't have to be one pathway. It is likely that all contribute, and the question really is what is the degree to which each element impacts outcomes. The idea of a shared contribution by each element should be added.
5. Note that the p38 effects depend on the isoform activation. If studies identify the specific isoforms activated by E2, please cite and explain if the findings are consistent (or not) with the known effects of the specific p38 isoforms.
6. One point that could be added which impacts outcomes is the lower rate at which females are referred to and complete cardiac rehabilitation.
7. Line 164. GPR30 activation was found to "cause cardioprotection".
8. Line 165. Should be "preserving".
9. Lines 321-323 refer to "one report" but two are cited.
10. Lines 336-339 don't make any sense. There is no conclusion to the "searching" action.
11. Line 503 strike the first "platelets".
N/A
Author Response
We would like to thank the Reviewer and appreciate the time and effort that have been taken to provide constructive comments on our work. All changes are highlighted in yellow in the manuscript. Please find below point-to-point responses:
1. Studies using ovariectomized animals provide insight into estrogen deficiencies, but do not recapture the more complex changes seen in human menopause. The VCD induced ovarian failure model (reviewed by Brooks HL et al Physiology. 31(4): 250-7. 2016) is likely a better model in terms of capturing more of these changes. Some acknowledgment in the review of the limitations of the OVX models and the potential of the VCD model would be helpful to put the OVX data in context.
We have now added a paragraph on the VCD model in section 2.2 Role of sex hormones
- The criticisms of estrogen replacement therapy are fair and correct, but a little too definitive. It should be noted that "some" studies show negative effects, instead of the more definitive presentation. The issue of timing does address some of these limitations and I appreciate the addition, but I believe the statement still requires some softening.
We have now rewritten this paragraph to milder statements about estrogen replacement efficacy, highlighting that negative, positive, and no change in effects on disease outcome have been observed in clinical studies.
- One area that is not addressed is the more complex effects driven by an androgen-estrogen imbalance after menopause. After menopause, androgen levels increase relative to estrogens (not necessarily an absolute increase). Because of the opposing effects of these hormones, some negative effects on the CV system may be a result of an imbalance in hormones. Some notation of this possibility could be added to the review (the idea that the negative effects are not just due to estrogen loss, but also due to an androgen:estrogen imbalance).
We have now added a paragraph on this phenomenon in section 2.2 Role of sex hormones
- Section 3 opens by pointing out that it is unclear if sex differences are driven by treatment disparities, comorbidities, or biological differences. But it doesn't have to be one pathway. It is likely that all contribute, and the question really is what is the degree to which each element impacts outcomes. The idea of a shared contribution by each element should be added.
We have added: ‘’It is likely that each of these factors contributes to sex differences in MI outcomes to a certain degree.’’
- Note that the p38 effects depend on the isoform activation. If studies identify the specific isoforms activated by E2, please cite and explain if the findings are consistent (or not) with the known effects of the specific p38 isoforms.
Only one study mentioned which p38 isoform was involved (p38beta), we have added a line referring to a study that shows that p38beta activation is indeed protective in cardiac ischemia, consistent with the studies on E2 and p38.
- One point that could be added which impacts outcomes is the lower rate at which females are referred to and complete cardiac rehabilitation.
We have now discussed sex differences in cardiac rehabilitation in section 3.3. Sex differences in standard of care
- Line 164. GPR30 activation was found to "cause cardioprotection".
Added
- Line 165. Should be "preserving".
Changed
- Lines 321-323 refer to "one report" but two are cited.
Removed ‘according to one report’
- Lines 336-339 don't make any sense. There is no conclusion to the "searching" action.
We have now re-written these lines
- Line 503 strike the first "platelets".
Done
Reviewer 2 Report
This manuscript looks at the sex differences across myocardial ischemia-reperfusion injury, examining the effectiveness of clinical therapies. Overall, the manuscript was very comprehensive and provided significant evidence in the review to illustrate the value of examining sex differences and how there is a major knowledge gap with regards to sex-based difference not only in regards to how the efficacy of therapeutics differs among sex but also how an approach could be taken to properly evaluate and consider sex across clinical trials to better improve therapeutic strategies. The manuscript is very well written and I would emphasize the importance of the table generated. This particular table should preferably fit on one page in order to be serve as a clear and concise comprehensive overview to the drugs discussed throughout the review.
Author Response
We would like to thank the Reviewer and appreciate the time and effort that have been taken to provide constructive comments on our work. All changes are highlighted in yellow in the manuscript.
As discussed with the Editorial team, upon acceptance of the manuscript, the journal's layout team will facilitate fitting the table on one page.
Reviewer 3 Report
​Controlla i dettagli 154 / 5.000Risultati della traduzione
Risultato di traduzione
In this review, the Authors have provided a precise and exhaustive explanation of the pathophysiological mechanisms involved in the sex-differences of IRI​. The bibliography is also appropriate and consistent with the specific issues covered. 154 / 5.000
Risultati della traduzione
Risultato di traduzione
In this review, the Authors have provided a precise and exhaustive picture of the pathophysiological mechanisms involved in the sex-differences of IRIMinor editing of english language required
Author Response
We would like to thank the Reviewer and appreciate the time and effort that have been taken to provide constructive comments on our work. All changes are highlighted in yellow in the manuscript. We have performed further language editing and spell check.